# Components of the LINC and NPC complexes coordinately target and translocate a virus into the nucleus to promote infection

**Chelsey C. Spriggs**[1]*, **Grace Cha**[1], **Jiaqian Li**[1,2], **Billy Tsai**[1]*

**1** Department of Cell and Developmental Biology, University of Michigan Medical School Ann Arbor, Michigan, United States of America, **2** Department of Biological Chemistry, University of Michigan Medical School Ann Arbor, Michigan, United States of America

* cspriggs@med.umich.edu (CCS); btsai@umich.edu (BT)

## Abstract

Nuclear entry represents the final and decisive infection step for most DNA viruses, although how this is accomplished by some viruses is unclear. Polyomavirus SV40 transports from the cell surface through the endosome, the endoplasmic reticulum, and the cytosol from where it enters the nucleus to cause infection. Here we elucidate the nuclear entry mechanism of SV40. Our results show that cytosol-localized SV40 is targeted to the nuclear envelope by directly engaging Nesprin-2 of the linker of nucleoskeleton and cytoskeleton (LINC) nuclear membrane complex. Additionally, we identify the NUP188 subunit of the nuclear pore complex (NPC) as a new Nesprin-2-interacting partner. This physical proximity positions the NPC to capture SV40 upon release from Nesprin-2, enabling the channel to facilitate nuclear translocation of the virus. Strikingly, SV40 disassembles during nuclear entry, generating a viral genome-VP1-VP3 subcomplex that efficiently crosses the NPC to enter the nucleus. Our results reveal how two major nuclear membrane protein complexes are exploited to promote targeting and translocation of a virus into the nucleus.

## Author summary

Although many DNA viruses cause infection by moving from the cell surface to the nucleus of the host cell, the molecular basis of this process remains enigmatic in many instances. Polyomavirus is a DNA tumor virus that is responsible for several debilitating human diseases, but how it reaches the nucleus to cause infection is mysterious. Here we discover that two major nuclear membrane protein complexes–LINC and NPC–act cooperatively to drive nuclear entry of SV40, the prototype polyomavirus. Illuminating a critical polyomavirus entry step may lead to novel strategies to combat polyomavirus-induced diseases.

**Data Availability Statement:** All relevant data are within the manuscript.

**Funding:** This work was supported by the National Institutes of Health https://www.nih.gov/ (grant

R01AI064296 to B. T. and grants F32GM133099
and K99GM141365 to C.C. S.). This work was also
supported by the Burroughs Wellcome Fund
Postdoctoral Enrichment Program https://www.
bwfund.org/ (PDEP 1021142 to C.C.S.). The
funders had no role in study design, data collection
and analysis, decision to publish, or preparation of
the manuscript.

**Competing interests:** The authors have declared
that no competing interests exist.

## Introduction

To cause infection, an incoming viral particle binds to a receptor on the plasma membrane of its target host cell [1]. The virus is then endocytosed and sorted along the complex endomembrane network to reach an intracellular destination that supports replication of the viral genome. This transport event is often orchestrated with disassembly of the viral particle to enable proper delivery of the virus into a distinct intracellular compartment. Despite this general model of viral entry, the mechanistic basis of these events, in many instances, is unclear. For example, how DNA viruses, such as the polyomavirus (PyV) family members, target to and translocate into the nucleus remain largely enigmatic. Developing a more coherent mechanistic understanding of these processes will identify new strategies to combat virus-induced diseases and provide novel understanding of nuclear transport mechanisms.

Upon receptor-mediated endocytosis [2], the prototypic PyV simian virus 40 (SV40) is delivered to the endosome [3]. The virus bypasses the Golgi apparatus and is instead directly transferred to the endoplasmic reticulum (ER) [4] via a mechanism that involves endosome-ER membrane contact sites [5]. In the ER lumen, redox-active PDI chaperones reduce and isomerize the inter-pentameric disulfide bonds of the VP1 coat protein [6, 7] and unfold the invading VP1 C-terminal arms [8]. The disruption of these forces destabilizes the virus and as a consequence, the hydrophobic VP2 and VP3 internal proteins of SV40 become exposed [8, 9]. This generates a hydrophobic particle that integrates into and penetrates the ER membrane at a discrete site called the ER-foci [10–14]. A cytosolic extraction machinery in turn ejects the virus in the ER-foci into the cytosol [12, 15–17].

Once SV40 escapes into the cytosol, it is further disassembled by a component of the dynein motor complex called bicaudal D2 (BICD2) [18]. BICD2 binds directly to cytosolic SV40 and disassembles the virus by triggering release of VP1 pentamers from the intact viral particle. This disassembled virus is subsequently delivered to the nuclear envelope—presumably by the dynein-BICD2 motor complex—from where the virus enters the nucleus. However, the mechanistic basis by which cytosol-localized SV40 is targeted to the nuclear envelope and how this step is coupled to nuclear translocation of the viral particle remains largely unclear. There is evidence that the nuclear localization signal (NLS) of SV40 VP2 and VP3, as well as the classic nuclear import machinery importin α, play a role in nuclear entry of SV40 [19–21]. Yet another study reported that the SV40 genome disassociates from the VP2 and VP3 capsids in the cytosol and enters the nucleus without them [22].

In this manuscript, we clarify the molecular basis of SV40 nuclear entry. Our results show that cytosol-localized SV40 is captured at the nuclear membrane via direct binding to Nesprin-2, a component of the linker of nucleoskeleton and cytoskeleton (LINC) nuclear membrane complex. Further, we identified the NUP188 subunit of the nuclear pore complex (NPC) as a new Nesprin-2-interacting partner. This positioning allows the NPC to engage SV40 when it is released from Nesprin-2 to promote nuclear translocation of the virus. During nuclear entry, SV40 undergoes a final disassembly step, presumably to enable its efficient transport across the size-dependent channel of the NPC. Overall, our data illuminate how two major nuclear membrane protein complexes are exploited to promote the nuclear targeting and translocation of a viral particle.

## Results

### Nesprin-2 is required for SV40 infection

To reach the nuclear envelope, we hypothesized that cytosol-localized SV40 is captured by a specific protein localized at the nuclear membrane. Because SV40 is partially disassembled in

the cytosol by the BICD2 adaptor of the trimeric dynein motor complex (composed of the dynein motor, dynactin, and an adaptor), we envisioned that the dynein-dynactin-BICD2 complex subsequently targets the virus to the nuclear membrane. As RanBP2, a subunit of the NPC that is localized to the cytosolic face of the nuclear membrane, and Nesprin-2, a component of the LINC nuclear membrane protein complex, have both been reported to bind to the dynein-dynactin-BICD2 complex [23, 24], we asked if RanBP2 or Nesprin-2 plays a role in SV40 infection.

To test this possibility, we used siRNA against either RanBP2, Nesprin-2, or a scrambled (Scram) control to individually deplete the proteins from CV-1 cells (used classically to study SV40 infection) (Fig 1A). Under the knockdown (KD) conditions, we assessed SV40 infection by monitoring expression of the virally-encoded large T antigen (TAg) in the host nucleus, which is used as a marker of successful nuclear arrival [25]. Using this approach, we found that SV40 infection was markedly impaired when Nesprin-2, but not RanBP2, was depleted from cells (Fig 1B), suggesting that Nesprin-2 plays a role in SV40 infection.

We then used a KD-rescue strategy to firmly establish a role for Nesprin-2 during SV40 infection. Structurally, full-length Nesprin-2 (N2G) is composed of an N-terminal calponin homology domain, followed by a long spectrin repeat domain, a motor (dynein/kinesin-1)-binding domain, a transmembrane domain, and a KASH (klarischt, ANC-1, Syne homology) domain that associates with SUN proteins in the perinuclear space (Fig 1C) [26, 27]. Because expression of full-length Nesprin-2 (approximately 800 kDa) is challenging, we expressed truncated versions of siRNA-resistant Nesprin-2 constructs under Nesprin-2 KD and evaluated whether SV40 infection could be restored. Specifically, cells transfected with the scrambled siRNA were co-transfected with the control plasmid FLAG- and HA-tagged GFP (FLAG-HA-GFP), while cells transfected with the Nesprin-2 siRNA were co-transfected with either FLAG-HA-GFP or truncated Nesprin-2 tagged with GFP (Nesprin-2-GFP) [27]. Following infection, we evaluated large TAg expression only in GFP-expressing cells.

Under Nesprin-2 KD conditions, expression of a Nesprin-2 construct lacking only the spectrin repeat domain (Fig 1C; miniN2G SR51-56 or construct #1 for simplicity) fully restored large TAg expression (Fig 1D), suggesting that this domain is dispensable for virus infection. This finding also demonstrates that the block in SV40 infection due to the Nesprin-2 siRNA is caused specifically by depletion of Nesprin-2 and not from unintended off-target effects. By contrast, expression of a Nesprin-2 construct lacking both the spectrin repeat and motor-binding domains (Fig 1C; miniN2G, construct #2) could not restore large TAg expression (Fig 1D), indicating that the motor-binding domain of Nesprin-2 is critical for supporting SV40 infection.

To determine if the calponin homology domain is necessary for virus infection, we expressed a construct that lacks the calponin homology (and spectrin repeat) domain (Fig 1C; SR51-56 KASH, construct #3) under Nesprin-2 KD and found that it also completely restored infection (Fig 1D). This shows that the Nesprin-2 calponin homology domain is not required for SV40 infection. Not surprisingly, expression of a construct in which the motor-binding domain is deleted from construct #3 (Fig 1C; SR52-56 KASH ΔAD, construct #4) could not restore SV40 infection under Nesprin-2 KD (Fig 1D), further supporting the idea that the Nesprin-2 motor-binding domain is important for SV40 infection. Because constructs #1–4 were properly expressed at the nuclear membrane (Fig 1E), the inability of construct #2 or #4 to rescue virus infection under Nesprin-2 KD cannot be due to their mislocalization. Taken together, our results strongly suggest that SV40 exploits the motor-binding region of Nesprin-2 to promote infection.

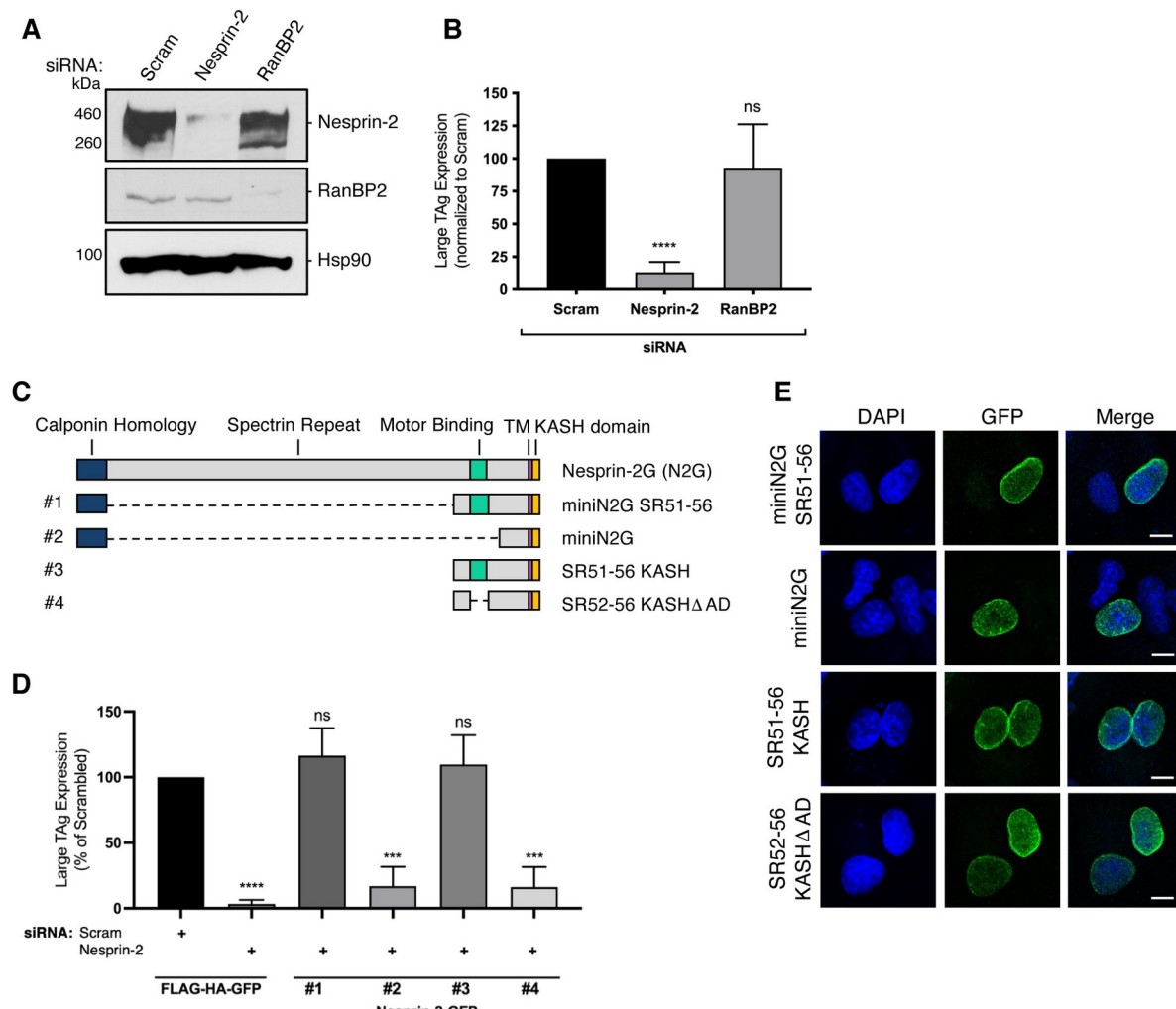

**Fig 1. Nesprin-2 is required for SV40 infection.** (A) CV-1 cells were transfected with 50 nM of either a scrambled control siRNA (Scram) or siRNA against Nesprin-2 or RanBP2 for 48 h. Protein levels were assessed by immunoblotting. Hsp90 was used as a loading control. (B) CV-1 cells were transfected with 50 nM of the indicated siRNA and infected with SV40 (MOI ~1). At 24 hours post infection (hpi), cells were fixed and stained for large T antigen (TAg). Data were normalized to the scrambled control. (C) Schematic of full-length Nesprin-2 and mutant constructs. (D) CV-1 cells were transfected with the scrambled control siRNA or siRNA against Nesprin-2 for 24 h. Cells were then transfected with either the FLAG-HA-GFP control construct or the indicated Nesprin-2-GFP constructs from 1C for an additional 24 h. Transfected cells were infected with SV40 (MOI ~1) for 24 h and fixed and stained to assess TAg as in 1B. Data were normalized to the scrambled control with FLAG-HA-GFP. (E) CV-1 cells were transfected with the indicated constructs for 48 h and fixed and stained for GFP (green). Cells were counterstained with DAPI (blue). Values are averages of the means (n = 3) ± SD. A standard Student's t test was used to determine statistical significance. **, P ≤ 0.005; ***, P ≤ 0.0005; ****, P ≤ 0.0001. Scale bars, 10μm.

## SV40 binds directly to Nesprin-2 during entry

To determine whether Nesprin-2 captures SV40 at the nuclear membrane to promote infection, we asked if Nesprin-2 binds to the virus during entry. To test this, endogenous Nesprin-2 was immunoprecipitated from SV40-infected CV-1 cells, and the precipitated material subjected to SDS-PAGE and immunoblotting. Our results showed that SV40 VP1 (Fig 2A), the viral genome, VP3, and a low level of VP2 (Fig 2B) all co-precipitated with Nesprin-2, while mock immunoprecipitation using a control (IgG) antibody did not. These findings indicate that Nesprin-2 associates with infectious (genome-containing) SV40 during entry.

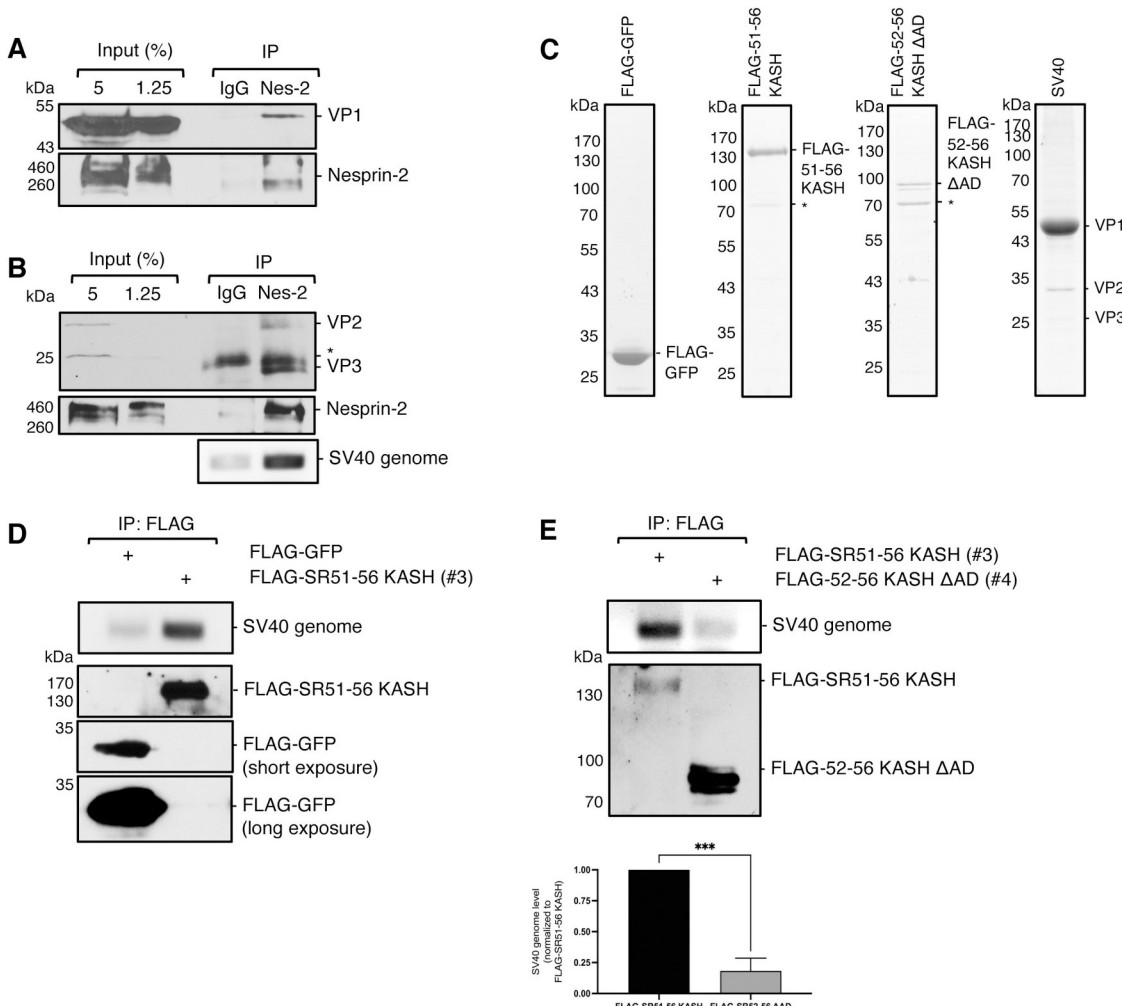

**Fig 2. SV40 binds directly to Nesprin-2 during entry.** (A) CV-1 cells were infected with SV40 (MOI ~25) for 20 h. Endogenous Nesprin-2 was immunoprecipitated (IP) from whole-cell extracts and eluted samples subjected to SDS-PAGE followed by immunoblotting for VP1. (B) As in A, except immunoprecipitation of Nesprin-2 was followed by immunoblotting for VP2/3. DNA was isolated from the eluted sample and PCR performed to identify SV40 genomic DNA. (C) SDS-PAGE of purified FLAG-GFP, FLAG-51-56 KASH, FLAG-52-56ΔAD, and SV40 visualized with Coomassie stain. (D) *In vitro* binding assay of SV40 with either FLAG-GFP or FLAG-51-56 KASH. Purified SV40 treated with low concentrations of DTT and EGTA was incubated with the indicated protein followed by incubation with anti-FLAG M2 magnetic beads. The immunoprecipitated material was eluted with excess 3x-FLAG and subjected to SDS-PAGE followed by immunoblotting with FLAG antibody. A fraction of immunoprecipitated material was taken for PCR analysis for the presence of SV40 genomic DNA. Western blots of FLAG-GFP showing a short and long exposure are included. (E) As in D, except SV40 was incubated with either FLAG-51-56 KASH or FLAG-52-56ΔAD. The bottom graph represents the quantification of SV40 genome levels of construct #3 and #4 normalized against the FLAG precipitated material (FLAG-SR51-56 KASH signal). Values are averages of the means (n = 3) ± SD. A standard Student's t test was used to determine statistical significance. ***, P ≤ 0.0005.

 To assess if SV40 binds directly to Nesprin-2, we used a purified *in vitro* binding system. A FLAG-tagged Nesprin-2 construct in which FLAG was appended to the N-terminus of construct #3 (FLAG-51-56 KASH) and a control FLAG-GFP construct were expressed in and subsequently purified from HEK 293T cells (Fig 2C). Purified SV40 (Fig 2C) was treated with the reductant DTT and calcium chelator EGTA to mimic conformational changes that the virus experiences in the ER lumen and upon arrival to the cytosol [6–8] and incubated with either FLAG-GFP or FLAG-51-56 KASH. The sample was then subjected to immunoprecipitation

with FLAG antibody-conjugated beads and PCR used to assess Nesprin-2 binding to the infectious virus. We found that precipitation of FLAG-51-56 KASH, but not FLAG-GFP, pulled down the viral genome (Fig 2D), indicating that infectious SV40 binds directly to Nesprin-2. We next asked if the Nesprin-2 motor-binding domain is required for SV40 binding by performing immunoprecipitation with either FLAG-51-56 KASH (construct #3) or FLAG-52-56 KASH ΔAD (construct #4) that lacks the motor-binding domain. Under these conditions, only FLAG-51-56 KASH was able to precipitate the viral genome (Fig 2E), suggesting that SV40 binds directly to Nesprin-2 through this region.

## Nesprin-2 promotes nuclear arrival of SV40

Because Nesprin-2 is a nuclear membrane protein, direct binding of cytosol-localized SV40 to Nesprin-2 would target the virus to the nuclear envelope, thereby initiating entry of the virus into the nucleus to cause infection. To test if Nesprin-2 is responsible for targeting SV40 to the nuclear membrane and its subsequent translocation into the nucleus, we used an imaging approach. By confocal microscopy, both SV40 VP1+ and VP2/3+ signals can be seen localized to Nesprin-2 (i.e. on the nuclear membrane; white arrows) and inside the nucleus (i.e. in the nucleoplasm; orange arrows) (Fig 3A and 3B). In these studies, the transcription inhibitor actinomycin D was added at the time of infection to ensure that only incoming viral particles, and not newly transcribed viral proteins, were tracked. To confirm that we were monitoring the fate of the infectious SV40 particle, ethynyl-2'-deoxyuridine (EdU)-labelled SV40 was used to determine whether the viral genome associates with the nuclear-localized capsid proteins. We found that in the nucleus, the VP1+ signal which colocalized with the VP2/3+ signal consistently colocalized with the EdU signal (Fig 3C, top row; white arrow), which is indicative of an infectious viral particle. However, a subset of cells (~30%) showed nuclear VP1 without the corresponding VP2/3+ or EdU signal (Fig 3C, bottom row; orange arrowhead), suggesting that a pool of VP1 can enter the nucleus without VP2/3 or the genome–these species likely represent non-infectious viral particles. For this reason, we monitored VP2/3 instead of VP1 to track the infectious viral particle in subsequent imaging experiments.

As expected, the Nesprin-2 siRNA (Fig 1) efficiently depleted Nesprin-2 expression by confocal microscopy (Fig 3D). Scrambled control and Nesprin-2 KD cells were then infected with SV40 and stained for VP2/3. In control cells, the VP2/3+ signal can be observed at the nuclear membrane and inside of the nucleus (Fig 3E, top row). By contrast, the VP2/3+ signal at the nuclear membrane and inside the nucleus was markedly decreased under Nesprin-2 KD (Fig 3E, bottom row). When quantified, Nesprin-2 depletion blocked the appearance of the VP2/3 + signal at the nuclear membrane by ~ 65% (Fig 3F) and inside the nucleus by ~75% (Fig 3G). These findings suggest that Nesprin-2 plays a critical role in targeting SV40 to the nuclear envelope for viral entry into the nucleus.

In Nesprin-2 KD cells, the VP2/3+ signal instead appears to accumulate as a discrete punctum that resembles the ER-foci—the site where SV40 penetrates the ER membrane to reach the cytosol and is disassembled by BICD2 [18, 28]. To determine whether the virus is trapped near the ER-foci under the Nesprin-2 KD condition, we co-stained these SV40-infected cells with BAP31, an ER marker that is recruited by the virus into the foci structure during ER escape [11]. BAP31 co-localized with the VP2/3+ signal under Nesprin-2 KD (Fig 3H) suggesting that SV40 is unable to transport from the ER-foci to the nuclear membrane in the absence of Nesprin-2.

To test whether SV40 is in the ER lumen or penetrated the ER-foci to reach the cytosol under Nesprin-2 KD conditions, we used a previously established sucrose gradient disassembly assay to monitor SV40 disassembly in the cytosol [28]. Using this assay, we found that

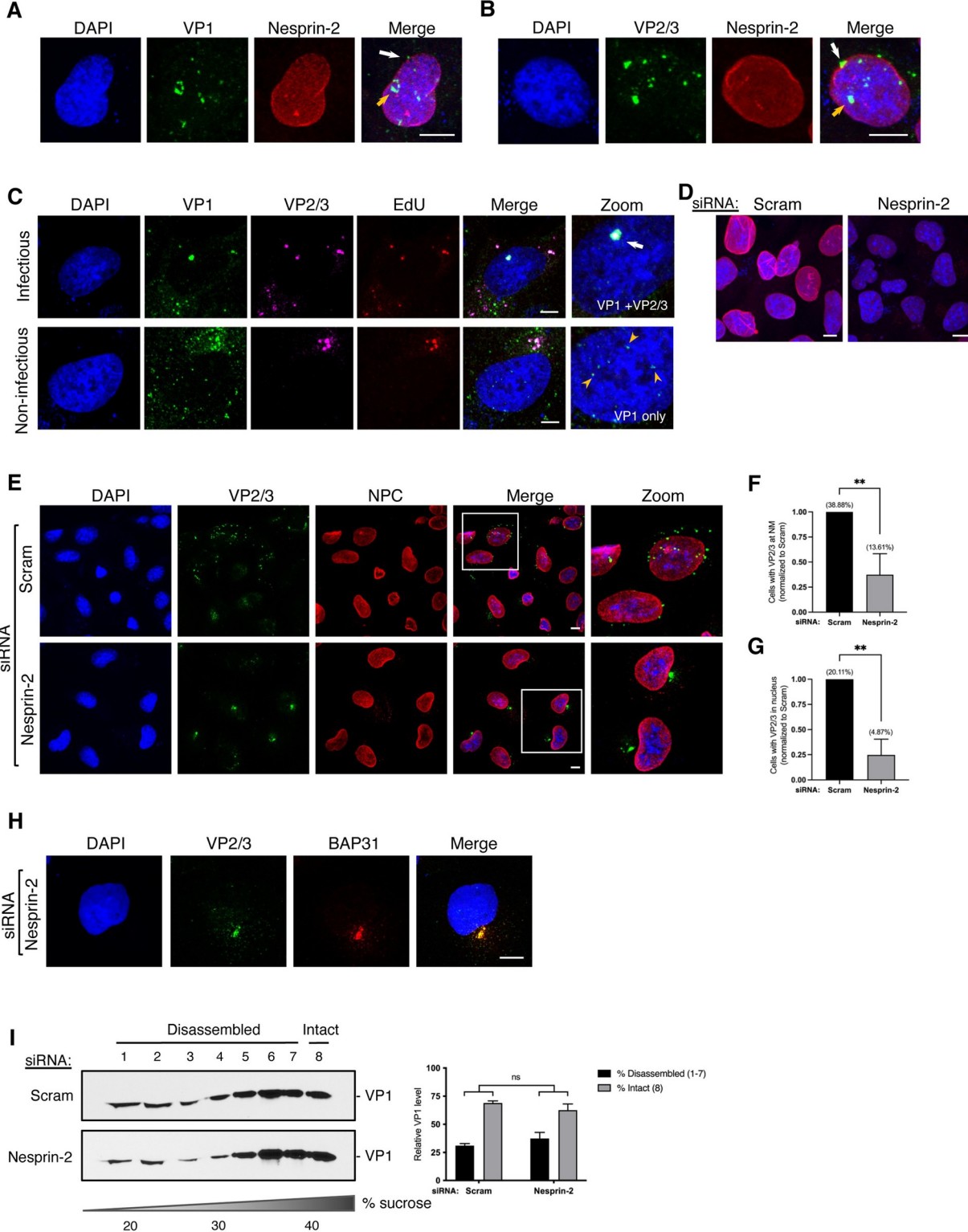

**Fig 3. Nesprin-2 promotes nuclear arrival of SV40.** (A) Confocal analysis of CV-1 cells that were infected with SV40 (MOI ~25) for 20 h in the presence of 1 μM actinomycin D (ActD) and stained with anti-VP1 (green) and anti-Nesprin-2 (red) antibodies. Cells were counterstained with DAPI (blue). White arrow = VP1 at nuclear membrane, orange arrow = VP1 inside the nucleus. (B) As in A, except cells were stained with anti-VP2/3 instead. White arrow = VP2/3 at nuclear membrane, orange arrow = VP2/3 inside the nucleus. (C) Confocal analysis of CV-1 cells infected with ethynyl-2'-deoxyuridine (EdU)-labelled SV40 (MOI ~25; red) for 20 h with 1 μM ActD. EdU labeling in cells was detected using

the Click-iT EdU reaction kit according to the manufacturer's protocol and then stained with anti-VP1 (green) and anti-VP2/3 (pink). Cells were counterstained with DAPI (blue). White arrow = VP1 with VP2/3 and genome, orange arrowhead = VP1 only (no genome) inside of the nucleus. (D) Confocal analysis of CV-1 cells that were transfected with the scrambled control siRNA or siRNA against Nesprin-2 for 48 h and stained with anti-Nesprin-2 (red) antibody. Cells were counterstained with DAPI (blue). (E) Confocal analysis of CV-1 cells that were infected with SV40 (MOI ~25) for 20 h with 1 μM ActD and stained with anti-VP2/3 (green) and anti-MAB414 that stains the NPC (red). Cells were counterstained with DAPI (blue). (F) Quantification of the percentage of cells in (E) that with VP2/3+ signal at the nuclear membrane normalized to the scrambled control. Actual values are shown in parenthesis. (G) As in F, except the percentage of cells with VP2/3+ signal inside of the nucleus is quantified. Actual values are shown in parenthesis. (H) Confocal analysis of CV-1 cells that were transfected with siRNA against Nesprin-2 for 48 h and then infected with SV40 (MOI ~25) for 20 h with 1 μM ActD. Cells were stained with anti-VP2/3 (green) and anti-BAP31 (red). Cells were counterstained with DAPI (blue). (I) CV-1 cells were transfected with either the scrambled control siRNA or siRNA against Nesprin-2 for 48 h, and infected with SV40 (MOI ~5). At 16 hpi, the cytosolic fraction was isolated, layered over a discontinuous sucrose gradient (20–40% sucrose), and centrifuged. Fractions were collected from the top of the gradient and the presence of SV40 (VP1) assessed by immunoblotting. The levels of VP1 in fractions 1–7 represent disassembled virus particles and fraction 8 contains intact virus. The graph represents the quantification of disassembled vs. intact virus. Values are averages of the means (n = 3) ± SD. A standard Student's t test was used to determine statistical significance. **, P ≤ 0.005. Scale bars, 10 μm.

SV40 disassembly in the cytosol (i.e. virus found in fractions 1–7 of the gradient) is unaffected by Nesprin-2 KD (Fig 3I; quantified in graph), indicating that the virus can successfully reach the cytosolic side of the ER-foci to undergo BICD2-dependent disassembly in the absence of Nesprin-2. These findings are consistent with the postulated function of Nesprin-2 in recruiting cytosol-localized virus to the nuclear membrane for entry.

## SV40 containing VP1 and VP3 transports into the nucleus

What viral components support transport of SV40 into the nucleus? Because the VP2/3 antibody used in our imaging analysis (Fig 3) recognizes both proteins, it cannot distinguish which viral proteins are present at the nuclear membrane or inside of the nucleus. To address this, we used a previously published biochemical fractionation assay to isolate the nuclear fraction from cells [29]. In this protocol (Fig 4A), cells are incubated with a low concentration of detergent (0.1% NP40) and centrifuged to generate a supernatant fraction that contains the non-nuclear cytoplasmic material (C) and a pellet fraction harboring the nuclear membrane and nucleoplasm (N). To verify the integrity of the fractionation procedure, the C and N fractions, along with the whole cell extract (W), were subjected to SDS-PAGE and immunoblotted with different organelle markers. Indeed, the nuclear marker histone H3 was found only in the N fraction, whereas the cytosolic (Hsp90), ER (BAP31), endosome (Rab7), plasma membrane ($Na^+/K^+$ ATPase), and lysosome (LAMP2) markers were found exclusively in the C fraction (Fig 4B). We used these compartment-specific markers because SV40 traffics from the plasma membrane, endosome, ER, cytosol, and then the nucleus to cause infection (Fig 4C). These findings establish the fidelity of the published nuclear isolation protocol used in our system.

CV-1 cells were then infected with SV40 and processed using this assay. Strikingly, we found that only VP3, but not VP2, appeared in the N fraction (Fig 4D, top panel, lanes 1–3); a pool of VP1 was also found in the N fraction (Fig 4D, bottom panel, lanes 1–3). These results indicate that VP3, in complex with a residual level of attached VP1 pentamers, delivers SV40 into the nucleus. Under Nesprin-2 KD conditions, the VP3 level in the N fraction decreased (Fig 4D, top panel, compare lane 6 to 3; quantified in graph on right), suggesting that appearance of VP3 in the N fraction is Nesprin-2-dependent. In contrast, we observed no significant decrease in the VP1 level in the N fraction under Nesprin-2 KD (Fig 4D, bottom panel, compare lane 6 to 3; quantified in graph on right). Although a pool of VP1 in complex with VP3 and viral genome is expected to reach the nucleus via a Nesprin-2-dependent pathway (Fig 3C, top panel), it is likely that a separate pool of VP1—uncoated from the core SV40 particle in the cytosol and devoid of VP3 and viral genome (Fig 3C, bottom panel)—uses its own NLS to reach the nucleus via a Nesprin-2-independent pathway. This would explain why the VP1 level

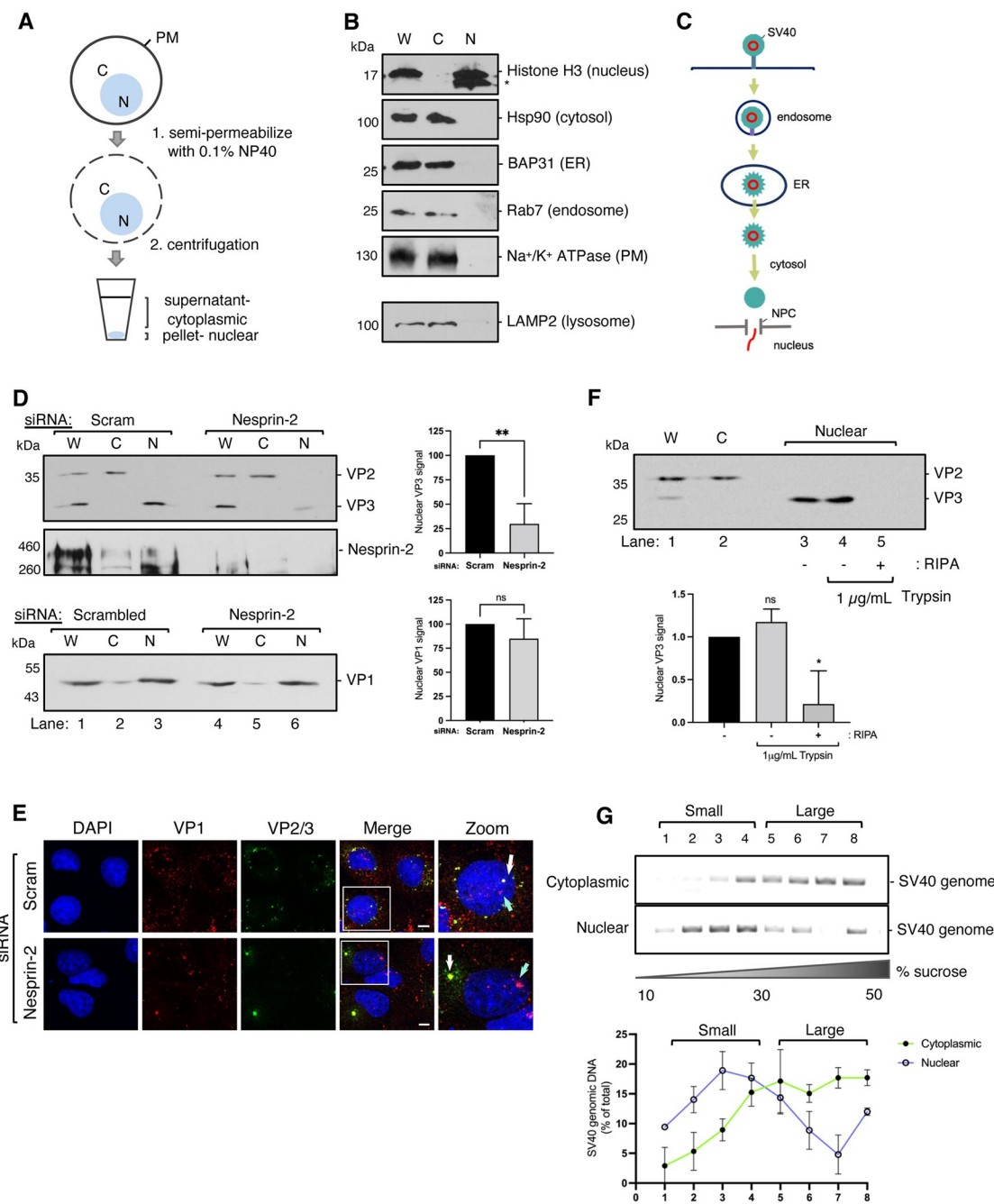

**Fig 4. SV40 containing VP1 and VP3 enters the nucleus.** (A) Schematic of the nuclear isolation protocol as described in [29]. (B) CV-1 cells were processed according to (A) and samples subjected to SDS-PAGE followed by immunoblotting with the indicated antibodies. W = whole cell lysate, C = cytoplasmic (non-nuclear) fraction, N = nuclear fraction. (C) Diagram of the organelles encountered by SV40 *en route* to the nucleus. (D) CV-1 cells were transfected with either the scrambled control siRNA or siRNA against Nesprin-2 for 48 h. Cells were then infected with SV40 (MOI ~5) for 20 h in the presence of 1 μM actinomycin D (ActD) and processed according to (A). Samples were subjected to SDS-PAGE followed by immunoblotting for VP2/3 (top), Nesprin-2 (middle) or VP1 (bottom). Graphs represent VP3 (top) and VP1 (bottom) levels in the nuclear fraction. (E) Confocal microscopy of CV-1 cells that were transfected with either the scrambled siRNA control or siRNA against Nesprin-2. Cells were infected with SV40 (MOI ~25) with 1 μM ActD and stained with anti-VP1 (red) and anti-VP2/3 (green). Cells were counterstained with DAPI (blue). White arrow = VP1 with VP3, light blue arrow = VP1 only. (F) CV-1 cells were infected with SV40 (MOI ~5) for 20 h and processed according to (A) except the nuclear fraction was then subjected to a protease protection assay in which lysates were treated with 1μg/mL trypsin in the presence or absence of detergent (RIPA) prior to SDS-PAGE and immunoblotting for VP2/3. The graph quantifies the nuclear VP3 levels in lanes 3,4 and 5. (G) CV-1 cells were infected with SV40 (MOI ~5) for 20 h with 1 μM

ActD and processed according to (A). Lysates were further solubilized with RIPA buffer and layered over a discontinuous sucrose gradient (10–50%). Following centrifugation, fractions were collected from the top of the gradient and the presence of SV40 genome detected by PCR. The graph represents the percentage of total SV40 DNA signal found in each fraction. Values are averages of the means (n = 3) ± SD. A standard Student's t test was used to determine statistical significance. $^{*}$, P $\leq$ 0.05; $^{**}$, P $\leq$ 0.005.

in the N fraction is largely unaffected when Nesprin-2 is depleted. To further evaluate this phenomenon, we again performed confocal microscopy on SV40-infected cells in the presence or absence of Nesprin-2. As expected, we found that while VP1 and VP3 co-localize inside the nucleus in control cells (Fig 4E, top panel, white arrow; see Fig 3C, top panel), VP3 is trapped at the ER-foci under Nesprin-2 KD (Fig 4E, bottom panel, white arrow; see Fig 3E, bottom panel). However, unlike VP3, some VP1 still gains entry into the nucleus in the presence or absence of Nesprin-2 (Fig 4E, blue arrows). These results are consistent with the finding that a pool of isolated VP1, without VP3 and the viral genome, can enter the nucleus (Fig 3C, bottom panel) and demonstrate that while nuclear VP3/genome remains in complex with VP1 to cause infection, some non-infectious VP1 can also gain entry to the nucleus independent of the rest of the complex.

To determine whether the VP3 observed in the N fraction is inside the nucleus or on the cytosolic surface of the nuclear membrane, the N fraction was subjected to limited proteolysis with trypsin. This protease protection assay revealed that the VP3 level did not decrease in the presence of trypsin unless the N fraction was treated with detergent (RIPA buffer) to disrupt the nuclear membrane (Fig 4F, compare lanes 3–5; quantified in the graph below). These findings indicate that VP3 in the N fraction is likely inside the nucleoplasm and not simply on the outer cytosolic face of the nuclear membrane. Taken together, our analysis suggests that the internal viral protein VP3, in association with VP1, drives SV40 transport into the nucleus.

## SV40 is further disassembled upon nuclear entry

Although SV40 undergoes BICD2-dependent disassembly in the cytosol, it is possible that the pore cargo size limit of the NPC (39 nm) still cannot accommodate the cytosol-localized viral particle for entry. To determine whether SV40 undergoes additional disassembly to cross the NPC, we analyzed the size of the infectious (genome-containing) virus in the cytoplasmic (C) and nuclear (N) fractions by sucrose gradient centrifugation analysis. The condition for the sucrose gradient of the cytosol disassembly assay (Fig 3I, 20%-40%) was adjusted to 10%-50% so that any small molecular weight viral species (generated when SV40 disassembles during cytosol-to-nuclear transport) can be readily observed. Strikingly, a large pool of infectious SV40 (viral genome) in the nucleus appeared in the lighter fractions (i.e. fractions 2–4) of the gradient corresponding to smaller disassembled virus, whereas most of the infectious virus in the cytosol were distributed in heavier fractions (i.e. fractions 4–8) reflecting association with a larger viral particle (Fig 3G; quantified in graph below). These results suggest that during cytosol-to-nuclear transport, SV40 undergoes further disassembly, presumably to efficiently enter the nucleus and cause infection.

## The NPC subunit Nup188 binds to Nesprin-2 and promotes SV40 nuclear entry

Our model thus far postulates that Nesprin-2 captures cytosol-localized SV40, thereby targeting the virus to the nuclear envelope. Upon release from Nesprin-2, we hypothesized that SV40 is translocated into the nucleus via the NPC. How this process is accomplished is unclear. A previous report suggested that importin α, a component of the classic nuclear import machinery [30], mediates SV40 nuclear entry [19]. In agreement with this, we found

that SV40 infection is blocked by importazole (Fig 5A), an inhibitor of importin β (which is an importin α-binding partner [31]) in a concentration-dependent manner. In addition, we previously performed an unbiased proteomics screen that identified several nucleoporins (NUPs)–building blocks of the NPC–as potential SV40 binding partners (Fig 5B) [32]. These data together support the idea that SV40 relies on the classic NPC-dependent nuclear import machinery for nuclear entry. To test if any of the identified NUPs play a role in SV40 infection, we knocked down the individual NUPs (Fig 5C) and found that depletion of many of them impaired virus infection, with NUP188 KD displaying the strongest phenotype (Fig 5D). Because depletion of NUP188 showed the most robust effect on SV40 infection, we asked whether it associates with the virus during entry. We found that immunoprecipitation of endogenous NUP188 does in fact pull down SV40 during infection (Fig 5E). Together, these data suggest that SV40 nuclear entry likely depends on many NUPs of the NPC, with NUP188 playing a key role.

These findings prompted us to test whether NUP188 might also interact with Nesprin-2. Immunoprecipitation of Nesprin-2 pulled down NUP188 (Fig 5F), indicating that NUP188 does indeed associate with Nesprin-2 in cells. BICD2 was also pulled down in this fraction, consistent with reports that it interacts with Nesprin-2 to target the dynein motor complex to the nuclear membrane [24]. Because NUP188 is in physical proximity to Nesprin-2, we envisioned a scenario in which Nesprin-2-bound SV40 is subsequently captured by the NUP188-containing NPC upon release from the LINC complex. In agreement with this, we found that KD of Nesprin-2 significantly blocked the ability of NUP188 to bind to the virus (Fig 5G; quantified in graph). As a control, we tested whether Nesprin-2 KD disrupts the ability of NUP188 to bind to other NUPs and found that NUP188 still associates with its established binding partner NUP93 (Fig 5H; quantified in graph [33]), indicating that NUP188 maintains its overall structural integrity in the absence of Nesprin-2. These findings suggest that Nesprin-2 acts upstream of NUP188 during SV40 entry by releasing the viral particle to the NPC to enable translocation of the virus into the nucleus.

Finally, we used confocal microscopy to further support a role for NUP188 in SV40 nuclear entry. As anticipated, the percentage of cells with VP2/3+ signal in the nucleus decreased under NUP188 KD (Fig 5I; orange arrow, quantified in Fig 5K), indicating that NUP188 is important for nuclear entry of the viral particle. Intriguingly, depletion of NUP188 also led to a decrease in the VP2/3+ signal at the nuclear membrane (Fig 5I; white arrows, quantified in Fig 5J). This is unexpected because SV40 should be trapped in the nuclear membrane without an intact NPC (i.e. NUP188 KD), suggesting that retention of SV40 at the nuclear membrane nonetheless requires an intact NPC. In all, our findings demonstrate that NUP188, as part of the NPC, acts in conjunction with Nesprin-2 of the LINC complex to drive nuclear translocation of SV40.

## Discussion

Nuclear transport is a poorly characterized step in the entry of many viruses, particularly amongst the DNA tumor virus family. This manuscript elucidates the nuclear entry mechanism of the DNA tumor virus SV40, the prototype polyomavirus. Previous studies suggested that the NLS in the SV40 structural proteins VP2 and VP3 act in concert with the nuclear import machinery importin α to promote nuclear entry of SV40 [19, 20]. Beyond these reports, deeper mechanistic understanding of cytosol-to-nuclear transport of SV40 was lacking.

Our data depict a model in which SV40 is targeted to the nuclear membrane by directly interacting with Nesprin-2 of the LINC nuclear membrane complex (Fig 6). In addition, we identify NUP188 of the NPC as a new Nesprin-2-interacting partner. This juxtaposition

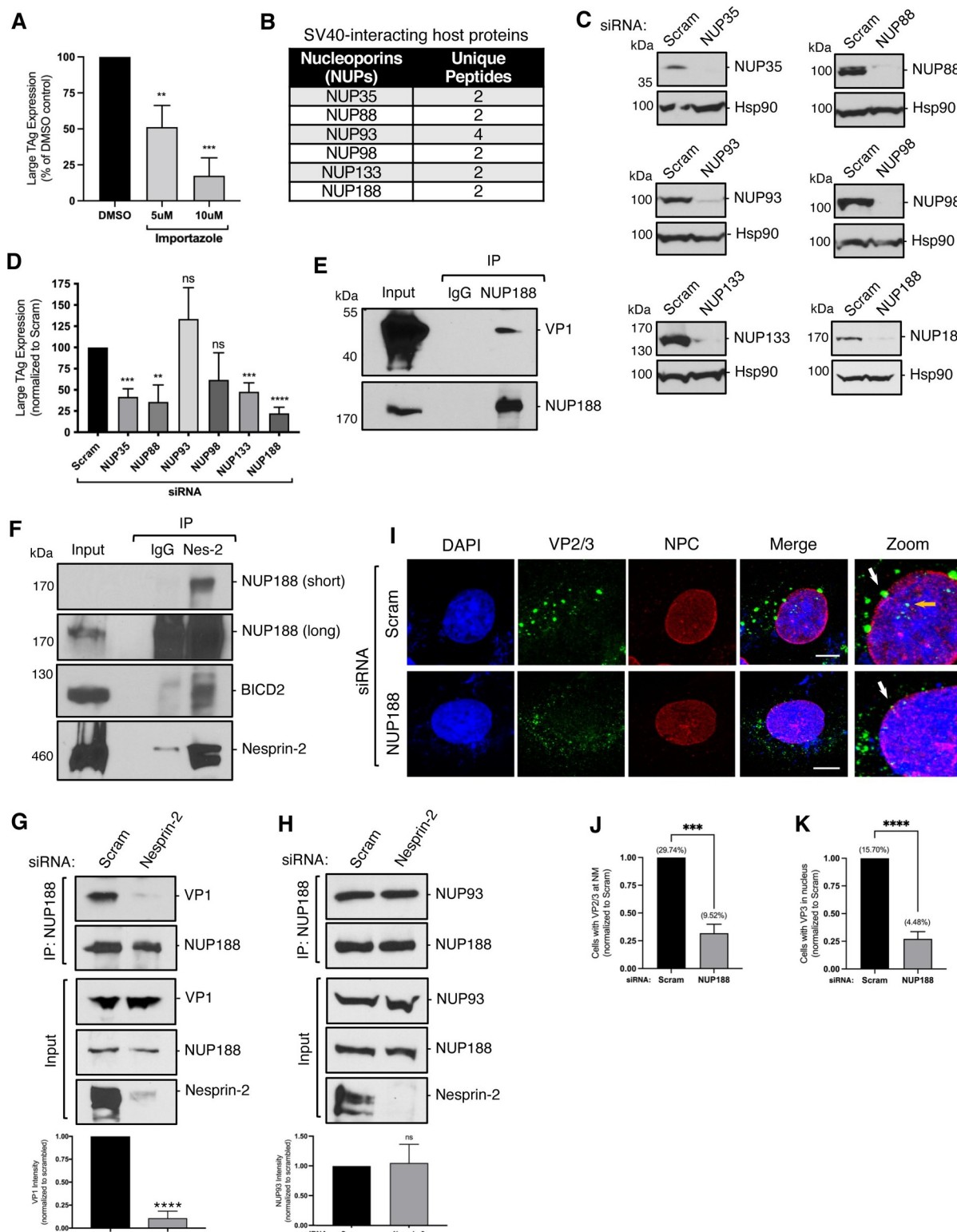

**Fig 5. The NPC subunit NUP188 binds to Nesprin-2 and promotes SV40 nuclear entry.** (A) CV-1 cells were infected with SV40 (MOI ~1) and treated with either a dimethyl sulfoxide (DMSO) control or importazole. At 24 hpi, cells were fixed and stained for large TAg. Data were normalized to the DMSO control. (B) Unique peptides corresponding to SV40-interacting host proteins identified by IP mass spectrometry [32]. (C) CV-1 cells were transfected with scrambled control siRNA, 50 nM siRNA against NUP35, NUP88, NUP93, NUP98, and NUP133, or 10 nM siRNA against NUP188 for 48 h. Protein levels were assessed by immunoblotting. Hsp90 was used as a loading control. (D) CV-1 cells

were transfected as in (C) and infected with SV40 (MOI ~1) for 24 h. Cells were fixed and stained for large TAg. Data were normalized to the scrambled control. (E) CV-1 cells were infected with SV40 (MOI ~25) for 20 h. Endogenous NUP188 was immunoprecipitated (IP) from whole-cell extracts and the eluted samples subjected to SDS-PAGE followed by immunoblotting for VP1. (F) CV-1 cells were infected with SV40 (MOI ~25) for 20 h. Endogenous Nesprin-2 was immunoprecipitated (IP) from whole-cell extracts and eluted samples subjected to SDS-PAGE followed by immunoblotting for NUP188 and BICD2. (G) CV-1 cells were transfected with either the scrambled control siRNA or siRNA against Nesprin-2 for 48 h. Cells were then infected with SV40 (MOI ~25) for 20 h and endogenous NUP188 immunoprecipitated from whole-cell extracts. The samples were subjected to SDS-PAGE followed by immunoblotting for VP1. (H) As in (G), except samples were immunoblotted for NUP93. (I) Confocal analysis of CV-1 cells that were transfected with either scramble control siRNA or siRNA against NUP188 for 48 h and then infected with SV40 (MOI ~25) for 20 h in the presence of 1 μM actinomycin D (ActD). Cells were stained with anti-VP2/3 (green) and anti-MAB414 that stains the NPC (red). Cells were counterstained with DAPI (blue). White arrow = VP2/3 at nuclear membrane, orange arrow = VP3 inside the nucleus. (J) Quantification of the percentage of cells in (I) with VP2/3 at the nuclear membrane normalized to the scrambled control. Actual values are shown in parenthesis. (K) As in J, except the percentage of cells with VP3 inside of the nucleus is quantified. Actual values are shown in parenthesis. Values are averages of the means (n = 3) ± SD. A standard Student's t test was used to determine statistical significance. **, P ≤ 0.005; ***, P ≤ 0.0005; ****, P ≤ 0.0001. Scale bars, 10 μm.

positions the NPC to capture SV40 upon release from Nesprin-2, enabling the channel to facilitate nuclear entry of the virus. Strikingly, SV40 undergoes further disassembly during cytosol-to-nuclear transport, producing a viral genome-VP1-VP3 subcomplex that translocates across the NPC to reach the nucleus. Thus, our results reveal how two major nuclear membrane protein complexes are hijacked to promote targeting and translocation of a virus into the nucleus.

During entry, SV40 is disassembled in the cytosol by the BICD2 dynein cargo adapter [18]. The disassembled virus is then transported to the nuclear membrane presumably by the BICD2-dynein motor complex. How the nuclear membrane captures the virus to initiate

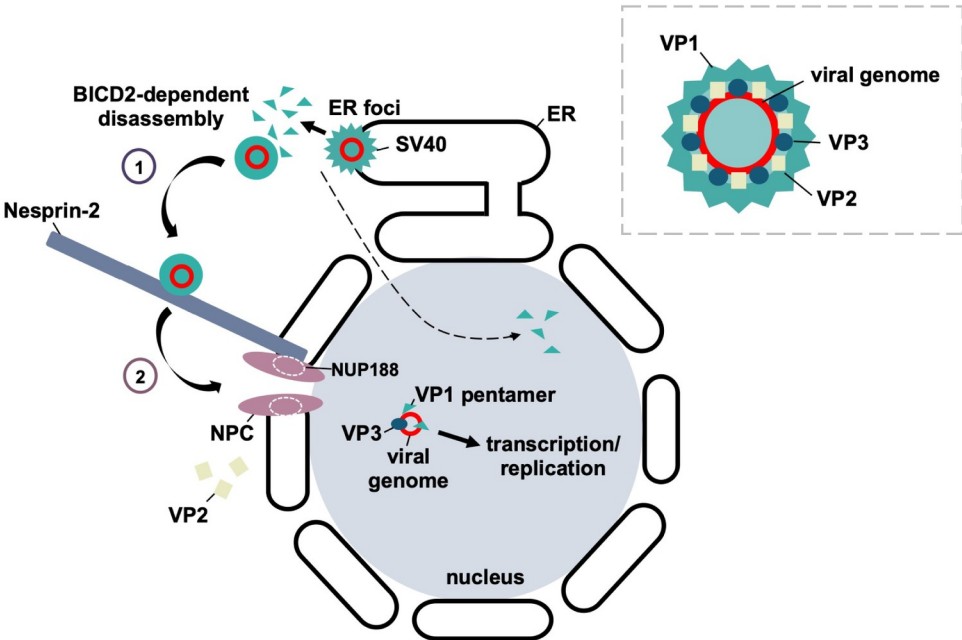

**Fig 6. Model.** Following virus escape from the endoplasmic reticulum (ER) at the viral-induced ER-foci and BICD2-dependent disassembly in the cytosol, SV40 is transported towards the nucleus where it is captured at the nuclear membrane through direct binding to Neprin-2 (step 1). Nesprin-2 is physically positioned proximal to the nuclear pore complex (NPC), in part, through its association with nucleoporin NUP188. This enables the successful transfer of SV40 from Nesprin-2 into the nucleoplasm through the NPC (step 2). During nuclear entry, the viral particle undergoes a final disassembly step such that the VP1 and VP3, but not VP2, capsid proteins accompany the viral genome into the nucleus to cause infection. The fate of VP2 is unknown. Inset: SV40 structural organization. The circular, double-stranded DNA viral genome is encased within a capsid shell composed of 72 pentamers of the major capsid protein VP1 (teal). Inside, each pentamer is associated with one copy of either the VP2 (beige) or VP3 (blue) minor capsid proteins.

nuclear entry remains unclear. Because Nesprin-2 [24] and RanBP2 [23] are both cytoplasmic-facing nuclear membrane proteins that bind to the BICD2-dynein motor complex, we hypothesized that one (or both) of these host factors may be responsible for capturing SV40 to facilitate nuclear entry and infection. Using siRNA KD, we found that Nesprin-2, but not RanBP2, plays an essential role during SV40 entry. Importantly, we identified the motor-binding domain of Nesprin-2 as critical for infection, consistent with the hypothesis that this domain is used to recruit SV40 to the nuclear membrane via the BICD2-dynein motor complex.

Previous studies have reported an essential role for RanBP2 (NUP358) in supporting nuclear translocation of other viruses [34, 35]. Here, we show the first instance of a virus using the LINC complex component Nesprin-2 to facilitate docking at the nuclear membrane instead. By co-immunoprecipitation, we established that Nesprin-2 engages infectious (genome-containing) SV40 during entry. *In vitro* binding assays further revealed that SV40 binds directly to Nesprin-2, and that this interaction is mediated largely by the motor-binding domain. These results are in agreement with our infection data, demonstrating a critical role of this domain during virus infection. Importantly, our data reveal that SV40 does not require BICD2 or cellular motor proteins to mediate its interaction with Nesprin-2 once the virus transports to the nuclear membrane. In fact, this finding raises the possibility that the motor-binding domain of Nesprin-2 may also function as a general cargo-binding domain for cellular substrates entering the nucleus.

Our confocal microscopy experiments revealed that depletion of Nesprin-2 markedly impaired localization of SV40 (VP2/3+ signal) to the nuclear membrane and into the nucleoplasm. Intriguingly, without Nesprin-2, cytosol-localized SV40 is not randomly diffused within the cytosol, but instead appears to concentrate on the cytosolic side of the ER-foci (where SV40 undergoes ER-to-cytosol transport and BICD2-dependent disassembly). This finding implies that the pathway leading SV40 from the cytosolic side of the ER-foci to the nuclear membrane may depend on Nesprin-2. Nesprin-2 is capable of recruiting both the dynein and kinesin-1 motors [24]. However, because both motors (and BICD2) are required for upstream steps in the SV40 entry pathway [32, 36], specific studies using carefully designed inducible systems would be required to capture the precise role of these proteins in transporting SV40 to the nuclear membrane.

Using a cell-based fractionation assay, we found that SV40 containing VP1 and VP3 (but not VP2) is transported into the nucleus. A limited protease digestion assay further demonstrated that VP3 is protected by an intact (detergent-sensitive) membrane, confirming that VP3 is in fact inside the nucleus. While we saw a decrease in VP3 levels inside of the nucleus in Nesprin-2 KD cells, we consistently observed no corresponding accumulation of VP3 in the cytoplasmic fraction. One explanation is that, during centrifugation, any VP3 that fails to reach the nuclear membrane becomes aggregated and therefore not detected in the cytoplasmic fraction in our system.

Interestingly, when Nesprin-2 initially captures SV40 on the cytosolic surface of the nuclear membrane, the viral particle contains VP1, VP3, as well as a low level of VP2 (Fig 2A and 2B). This suggests that cytosol-localized SV40 experiences a second disassembly step during cytosol-to-nuclear transport such that VP2 is released from the core viral particle prior to nuclear entry. We used a sucrose sedimentation assay to test this possibility and found the formation of small molecular weight SV40 in the nucleus but not the cytosol. These results are consistent with the idea that SV40 undergoes additional disassembly during cytosol-to-nuclear transport. We hypothesize that this disassembly event—generating SV40 with VP1, VP3, and genome–produces a reduced-size viral particle that can more readily cross the nuclear membrane to cause infection. This may be the result of either VP2-associated VP1 pentamers being removed from the core viral particle or additional conformational changes that occur during virus

translocation across the NPC. Although the force required to disassemble SV40 at the nuclear membrane is unclear, the motor activity of kinesin-1 has been reported to facilitate disassembly of adenovirus at the nuclear membrane [37]. Whether kinesin-1 acts in a similar capacity to trigger disassembly of SV40, as well as the precise physical modifications responsible for the change in mobility (through the sucrose gradient), remain to be investigated.

VP2 and VP3 are produced through alternative splicing where the entire VP3 sequence is contained within VP2. As both proteins contain an NLS, it has been assumed that they serve largely redundant functions during viral entry. What then differentiates them such that VP3 enters the nucleus while VP2 does not? As the genome with VP3 (and VP1) often moves into a larger, distinct foci inside of the nucleoplasm (Figs 3C and 4E), it is possible that the viral capsids play a role in directing the genome to a specific location inside of the nucleus post nuclear arrival, reminiscent of the murine PyV replication center in the host nucleus [38–41]. Identifying a role for VP3 in downstream nuclear events may illuminate important distinctions between VP3 and VP2 that explain the observed divergence in their fates during entry.

Previous studies showed that the NLS of SV40 VP3, along with importin α, is important for SV40 infection [19], suggesting that the NPC functions as the protein-conduit during nuclear entry. Using an unbiased proteomics approach, we identified several NUPs (Fig 5B) as potential virus-binding partners [32]. Here, we individually depleted these proteins and found that KD of most of them, but particularly NUP188, blocked SV40 infection. NUP188 also binds to SV40, and depletion of this NUP impaired nuclear entry of the virus. These findings strongly support the model that the NPC promotes the nuclear entry of SV40, with NUP188 playing a key role. Importantly, we found that NUP188 associates with Nesprin-2 (by IP), thereby positioning Nesprin-2 proximal to the NPC. We postulate that this physical proximity enables SV40 to be released from Nesprin-2 and efficiently transferred to the NPC for entry into the nucleus.

Knockdown of Nesprin-2 leads to a decreased interaction between SV40 and NUP188 (Fig 5G), indicating that Nesprin-2 acts upstream of NUP188 during viral infection. In this case, we expect to see a buildup of virus at the nuclear membrane when NUP188 is knocked down causing its path through the NPC to be obstructed. Instead, our imaging data revealed a decrease in SV40 (VP2/3+ signal) at the nuclear membrane in NUP188 KD cells. This indicates that the virus-Nesprin-2 interaction is likely weak and/or transient so that when Nesprin-2 cannot properly release SV40 to the NPC (due to a compromised NPC), it cannot be retained at the nuclear membrane; the fate of SV40 when NUP188 is depleted remains unknown. In sum, our data reveal how two nuclear membrane protein complexes coordinate to facilitate the nuclear entry of a DNA tumor virus. Future structural studies on the relationship between the NPC and LINC complexes are needed to further clarify their role in coordinating both viral and cellular nuclear import.

## Materials and methods

### Cell lines and reagents

CV-1 and HEK 293T cells were obtained from ATCC. Cells were grown in complete Dulbecco's modified Eagle's medium (DMEM) containing 10% fetal bovine serum, 10 U/mL penicillin, and 10 μg/mL streptomycin (Gibco, Grand Island, NY). DMEM, Opti-MEM and 0.25% trypsin-EDTA were purchased from Invitrogen (Carlsbad, CA). Bovine Serum Albumin (BSA), dithiothreitol (DTT), actinomycin D and importazole were purchased from Millipore Sigma (St Louis, MO). Phenylmethylsulfonyl fluoride (PMSF) was purchased from Thermo Scientific (Waltham, MA).

## Preparation of SV40

SV40 was prepared using the OptiPrep gradient system (Sigma) as previously described [28]. In brief, cells transfected with the SV40 viral genome were lysed in HN buffer (50 mM HEPES (pH 7.5), 150 mM NaCl) with 0.5% Brij58 for 30 min on ice. Following centrifugation, the supernatant was layered on top of a discontinuous 20% and 40% OptiPrep gradient. Tubes were centrifuged at 49,500 rpm for 2 h at 4˚C in a SW55Ti rotor (Beckman Coulter, Indianapolis, IN). Purified virus was collected from the white interface that forms between the OptiPrep layers and the aliquots stored at -80˚C for future use. For 5-ethynyl-2'-deoxyuridine (EdU)-labelling of SV40, 50 μM EdU was added to growth media during each round of virus propagation for incorporation into the infectious viral particle. Cells were later infected with SV40-EdU and the Click-iT EdU Imaging Kit (Thermo Scientific) was used for detection of the viral genome.

## Antibodies

SV40 large T antigen (Santa Cruz sc-147), monoclonal VP1 was provided by Dr. Walter Scott (University of Miami), polyclonal VP1 (Abcam ab53977), SV40 VP2/3 (Abcam ab53983), RanBP2 (Thermo Scientific A301-796A), Nesprin-2 (Bethyl A305-393A, Thermo Scientific MA5-18075), Hsp90 (Santa Cruz sc-13119), GFP (Dawen Cai, University of Michigan), FLAG (Millipore Sigma F7425), MAB414 NPC (Abcam ab24609), Histone H3 (Cell Signaling 9715S), BAP31 (Pierce MA3-002), Rab7 (Millipore Sigma R8779), Na+/K+ ATPase (Abcam ab76020), LAMP2 (Abcam ab18529), NUP35 (Millipore Sigma HPA018410), NUP88 (Santa Cruz sc-365868), NUP93 (Thermo Scientific A303-979A), NUP98 (Cell Signaling 2598S), NUP133 (Santa Cruz sc-376763), and NUP188 (Thermo Scientific PA5-66645), BICD2 (Abcam (ab117818).

## Plasmids

FLAG-HA-GFP was a gift from Wade Harper (AddGene plasmid #22612; http://n2t.net/addgene:22612; RRID: Addgene_22612). Nesprin-2 constructs (miniN2G, miniN2G SR51-56, SR51-56 KASH, and SR52-56 KASHΔAD) were a gift from Dr. Gregg Gunderson (Columbia University). pCMV-FLAG-SR51-56 KASH and pCMV-FLAG- SR52-56 KASHΔAD were generated for this study according to [42].

## siRNA transfection

All Star Negative (Qiagen, Valencia, CA) was used as a scrambled control siRNA. The following siRNAs were used in this study: siNesprin-2: GAGCAUCACUACAAGCAAAUG, siRanBP2: CGAAACAGCUGUCAAGAAA, siNUP35: UGCCCAGUUCUUACCUGGA, siNUP88: GGAAAUGGCUGAGCGUUUA, siNUP93:AGAGUGAAGUGGCAGACAA, siNUP98: GUGAAGGGCUAAAUAGGAA, siNUP133, siNUP188: GUAGAGAACUGUGG ACUAU. For knockdown experiments, CV-1 cells were reverse transfected with either 50 nM or 10 nM of the indicated siRNA using Lipofectamine RNAiMAX (Invitrogen). Infections and biochemical assays were all performed at 24–48 h post transfection.

## DNA transfection

For CV-1 cells, plasmids were transfected into 50% confluent cells using the FuGENE HD (Promega, Madison, WI) transfection reagent. DNA was allowed to express for at least 24 h prior to experimentation. For 293T cells, polyethylenimine (PEI; Polysciences, Warrington, PA) was used. DNA was allowed to express for at least 24 h prior to experimentation.

## Immunofluorescence and confocal microscopy

Cells were grown on no.1 glass coverslips and fixed with 1% formaldehyde for 15 min. Cells were then permeabilized in PBS with 0.2% Triton X-100 for 5 min and blocked with 5% milk containing 0.02% Tween-20. Primary antibodies were incubated in milk for 1 h at room temperature or overnight at 4˚ C. Coverslips were then washed 3x in milk and incubated with Alexa Fluor secondary antibodies (Invitrogen) for 30 min at room temperature. Coverslips were again washed and mounted using ProLong Gold with DAPI (Invitrogen). Images were taken on either a Nikon Eclipse TE2000-E inverted epifluorescence microscope or a Zeiss LSM 780 confocal laser scanning microscope. FIJI software was used for image processing and analysis. For each experiment, Z-stack images were taken and at least 100 cells were quantified as either positive or negative for VP2/3+ signal at the nuclear membrane (discrete signal colocalized with the MAB414 antibody) or in the nucleus (discrete signal within the border of the nuclear membrane). At least three independent replicates were quantified for each experiment and a standard Student's t test was used to determine statistical significance.

## Immunoprecipitation

Cells were lysed in RIPA buffer (50 mM Tris pH 7.4, 150 mM NaCl, 0.25% sodium deoxycholate, 1% NP40, 1 mM EDTA) with 1 mM PMSF for 15 min on ice. Cells were then centrifuged at 13,000 rpm for 10 min at 4˚ C. Input (5% and 1.25%) was collected from the resulting supernatant before 2 μg of the indicated antibody was added to the lysate and rotated overnight at 4˚ C. The next day, either Protein A/G agarose beads (Pierce) or Protein G Dynabeads (Thermo Scientific) were washed 3x with RIPA buffer and rotated with the lysate at 4˚ C for 30 min or 1 h, respectively. The tubes were then centrifuged at 7,000 rpm for 1 min or collected on a magnet and the beads washed 3x with RIPA buffer. For genome extraction, 40 μL of RIPA with 1% SDS was added to the beads and vortexed at room temperature for 5 min. Of this, 20% (8 μL) was removed, 40 μL of PB Buffer (Qiagen) added and samples run through a Qiagen column. DNA was extracted according to the miniprep protocol. SDS sample buffer was added to the remaining IP lysate and the beads boiled for 10 min at 95˚ C. Samples were then run on an SDS-PAGE gel and protein levels determined by immunoblotting.

## *In vitro* binding assay

Protein purification: HEK 293T cells were transfected with FLAG-GFP, FLAG-SR51-56 KASH or FLAG-SR52-56 KASHΔAD and lysed in RIPA buffer. Samples were centrifuged at 13,000 rpm and the supernatant subjected to FLAG immunoprecipitation using anti-FLAG M2 affinity gel (Millipore Sigma A2220) for 2 h at 4˚C. The samples were then washed with RIPA containing 0.1% NP40 and the proteins eluted using excess 3xFLAG peptide. Binding assay: Purified SV40 was treated with DTT and EGTA (400 μM) for 10 min at RT and incubated in HN buffer (50 mM HEPES and 150 mM NaCl in PBS) with either purified FLAG-GFP, FLAG-SR51-56 KASH or FLAG-SR52-56 KASHΔAD for 20 min at 37˚C with shaking. Lysates were immunoprecipitated using anti-FLAG M2 magnetic beads (Millipore Sigma M8823) for 20 min at RT and washed with 0.1% containing RIPA. Proteins were eluted with excess 3xFLAG peptide for 30 min at RT. A fraction of the eluant was removed for PCR of the viral genome and the remainder subjected to SDS-PAGE and immunoblotting.

## SV40 disassembly assay

Cytosolic or nuclear fractions were layered over a 20-30-40% or 10-30-50% discontinuous sucrose gradient. Tubes were centrifuged for 30 min at 4˚ C at 50,000 rpm. From the top,

25 µL aliquots were collected and subjected to SDS-PAGE followed by immunoblotting or PCR for the viral genome.

## PCR of viral genome

SV40 genomic DNA was amplified using the KOD Hot Start DNA polymerase (Millipore-Sigma) with primers against the viral genome (Fwd, 5′-GCAGTAGCAATCAACCCACA-3′; Rev, 5′-CTGACTTTGGAGGCTTCTGG-3′). Amplified DNA was subsequently run on an agarose gel to detect viral DNA.

## Nuclear Fractionation

Fractionation was performed according to a previously published rapid, efficient and practical (REAP) protocol for nuclear fractionation with minor modification [29]. Briefly, cells were resuspended in PBS + 0.1% NP40 and 1/3 collected as whole cell lysate. The remaining sample was centrifuged at 13,000 rpm for 10 min at 4°C. The resulting supernatant was collected as the cytoplasmic fraction. The pellet was washed in PBS + 0.1% with 500 mN NaCl and centrifuged at 13,000 rpm. The pellet was then resuspended in PBS + 0.1% NP40 to obtain the nuclear fraction.

## Protease protection

Following fractionation, the nuclear pellet was divided and resuspended in either PBS or RIPA buffer for 5 min. 1 µg/mL trypsin was added to each sample and incubated with shaking for 10 min at 25°C. Reactions were then quenched with 10% trichloroacetic acid (TCA) on ice for 10 min. Samples were spun at 13,000 rpm for 10 min and supernatant discarded. The resulting pellets were resuspended in SDS sample buffer with 2 M Tris pH 8.8, boiled at 95°C and subjected to SDS-PAGE followed by immunoblotting.

## Quantification of western blots

All western blots were developed on film and quantified using the FIJI software. For disassembly experiments, the signal from disassembled virus (fractions 1–7) was compared to that of intact virus (fraction 8) normalized to the total signal (fractions 1–8). For fractionation experiments, nuclear VP1 or VP3 signals were normalized to the respective levels of VP1 or VP3 in the whole cell lysate. For IP experiments, signals were normalized to the total immunoprecipitated material. At least three independent replicates were quantified for each experiment and a standard Student's t test was used to determine statistical significance.

## Acknowledgments

We thank members of the Tsai laboratory for thoughtful discussions throughout this work and the Cai laboratory at the University of Michigan for sharing its microscope facility.

## Author Contributions

**Conceptualization:** Chelsey C. Spriggs, Billy Tsai.

**Data curation:** Chelsey C. Spriggs.

**Formal analysis:** Chelsey C. Spriggs.

**Funding acquisition:** Chelsey C. Spriggs, Billy Tsai.

**Investigation:** Chelsey C. Spriggs, Grace Cha, Jiaqian Li.

**Methodology:** Chelsey C. Spriggs, Billy Tsai.

**Project administration:** Billy Tsai.

**Supervision:** Billy Tsai.

**Validation:** Chelsey C. Spriggs, Billy Tsai.

**Visualization:** Chelsey C. Spriggs, Billy Tsai.

**Writing – original draft:** Chelsey C. Spriggs, Billy Tsai.

**Writing – review & editing:** Chelsey C. Spriggs, Grace Cha, Billy Tsai.

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
