## [Decision Letter · Decision Letter 0]

9 Aug 2022

Dear Dr. Spriggs,

Thank you very much for submitting your manuscript "Components of the LINC and NPC complexes coordinately target and translocate a virus into the nucleus to promote infection" for consideration at PLOS Pathogens. As with all papers reviewed by the journal, your manuscript was reviewed by members of the editorial board and by several independent reviewers. The reviewers appreciated the attention to an important topic. Based on the reviews, we are likely to accept this manuscript for publication, providing that you modify the manuscript according to the review recommendations.

Sincerely,

Walter J. Atwood

Associate Editor

PLOS Pathogens

Karl Münger

Section Editor

PLOS Pathogens

Kasturi Haldar

Editor-in-Chief

PLOS Pathogens

orcid.org/0000-0001-5065-158X

Michael Malim

Editor-in-Chief

PLOS Pathogens

orcid.org/0000-0002-7699-2064

Reviewer Comments (if any, and for reference):

Reviewer's Responses to Questions

**Part I - Summary**

Reviewer #1: Spriggs et al. report the mechanism of nuclear translocation by SV40, the prototype polyomavirus (PyV), through engagement of nuclear membrane proteins, Nesprin-2 of LINC and NUP188 of the NPC. Additionally, this work demonstrates that virus disassembly steps occur during nuclear entry, and a subcomplex of viral genome and capsid proteins VP1 and VP3 cross the NPC to mediate delivery of the viral genome to the nucleus. This work builds upon previous findings that SV40 PyV is delivered to the nuclear envelope, possibly by a dynein-BICD2 complex. Overall, the findings in this paper significantly advance the field of viral entry and trafficking, the experiments are elegantly designed, the data are compelling, and the paper is well written.

Reviewer #2: This excellent paper by Spriggs et al. investigates the biology of SV40 nuclear import, identifying and mapping a novel interaction between SV40 and the motor binding domain of Nesprin-2 for Nup188-dependent nuclear import of VP3/genome. These aspects of SV40 (and polyomavirus) biology were not well defined and using coIPs, MS/proteomics, microscopy, and some customized novel assays, this paper does an excellent job to identify new important interactions that illuminate this area of the SV40 life cycle. The data are very sound and well controlled, the science is solid, and the paper is very well written. Only a few minor concerns are outlined below.

**Part II – Major Issues: Key Experiments Required for Acceptance**

Reviewer #1: No major issues

Reviewer #2: No major issues. Overall a fantastic paper that advances the field of nuclear entry of viruses.

**Part III – Minor Issues: Editorial and Data Presentation Modifications**

Reviewer #1: - Abstract, Line 26: change to “for most DNA viruses…”

- Fig 2 – Is the partial disassembly of the virus by treatment w/ DTT/EGTA established in the literature? Can you provide a reference on line 167?

- Figure 2 legend – D is not described. Maybe C on line 491 should be D? Also, the short and long exposure are not described.

- The PCR analysis performed in Figure 2 is not described in the M&Ms and should be included.

- It’s unclear why Fig. 2E has significantly reduced FLAG-SR51-56 KASH in comparison to the deltaAD or in D? Please provide an explanation or consider adding quantitation of several replicates.

- Is it possible that the non-infectious VP1 particles that enter the cell are intentional by-products of replication that have a pro-viral effect on SV40 infection or are the empty particles an effect of using high MOIs in these studies?

- Please include more detail on how the images and W blots were quantified in Fiji.

- Figure 5G – The input Nup188 protein levels appear to be reduced in Nesprin-2-treated cells. Given that Nup188 interacts with Nesprin-2, is KD of Nesprin-2 reducing Nup188? Can this be quantified to demonstrate if input is not reduced or approached in another way possibly with an additional control KD such as RanBP2 KD? An explanation could be provided in the Discussion.

Reviewer #2: Line 280/281. If VP1 enters the nucleus independently of Nesprin-2 and VP3 (as shown in Figure 4D) then why is it described as in complex with VP3/genome? Seems it wouldn’t be in complex right? This was confusing.

The disassembly data in 5G- genome was detected but what’s responsible for the change in mobility through the sucrose gradient? Dissociation of VP2 or VP3 from genome? Loss of VP1 pentamers? Loss of histones? Although some blots could clarify the speculation/insight provided in the discussion, this is merely a suggestion to potentially improve the paper- I would not consider this a mandatory requirement for acceptance as it can be addressed in future studies.

Nup188 silencing resulted in the strongest inhibition of SV40 infection, but other Nups were close. Do these coIP with the Nestrin-2/SV40 comlex too? Does Nestrin-2 silencing affect the interaction between these Nups and SV40? Again, this could be explored in future work.

What component of SV40 actually binds Nestrin-2? VP3? VP1? If this could be easily clarified it would strengthen the paper, if not then can be addressed in future work.

The final model figure could be improved to show more details concerning VP1, VP2, VP3, etc. What is the teal aura around the viral genome represent? VP1 pentamers as a capsid?

The EdU-labeling methodology of SV40 needs to be better detailed in Materials and Methods. What concentration of EdU was used, when was it added post-infection, etc.

PLOS authors have the option to publish the peer review history of their article (what does this mean?). If published, this will include your full peer review and any attached files.

Reviewer #1: No

Reviewer #2: **Yes: **Samuel K. Campos

Figure Files:

Data Requirements:

Reproducibility:

References:

---

## [Editor Report · Decision Letter 1]

22 Aug 2022

Dear Dr. Spriggs,

We are pleased to inform you that your manuscript 'Components of the LINC and NPC complexes coordinately target and translocate a virus into the nucleus to promote infection' has been provisionally accepted for publication in PLOS Pathogens.

Best regards,

Walter J. Atwood

Associate Editor

PLOS Pathogens

Karl Münger

Section Editor

PLOS Pathogens

Kasturi Haldar

Editor-in-Chief

PLOS Pathogens

orcid.org/0000-0001-5065-158X

Michael Malim

Editor-in-Chief

PLOS Pathogens

orcid.org/0000-0002-7699-2064
---

## [Editor Report · Acceptance letter]

31 Aug 2022

Dear Dr. Spriggs,

We are delighted to inform you that your manuscript, "Components of the LINC and NPC complexes coordinately target and translocate a virus into the nucleus to promote infection," has been formally accepted for publication in PLOS Pathogens.

Best regards,

Kasturi Haldar

Editor-in-Chief

PLOS Pathogens

orcid.org/0000-0001-5065-158X

Michael Malim

Editor-in-Chief

PLOS Pathogens

orcid.org/0000-0002-7699-2064